# Research on High Power Factor Single Tube Variable Structure Wireless Power Transmission

**Yi Yang [1,2,\*], Xuejian Zhang [1], Lei Luo [1], Shiyun Xie [1,2] and Qingshan Zhou [1]**

1   School of Electrical and Electronic Engineering, Chongqing University of Technology, Chongqing 400054, China; zhangxuejian@2019.cqut.edu.cn (X.Z.); luolei@2019.cqut.edu.cn (L.L.); xieshiyun1987@cqut.edu.cn (S.X.); zhouqingshan@2019.cqut.edu.cn (Q.Z.)
2   Chongqing Energy Internet Engineering Technology Research Center, Chongqing 400054, China
*   Correspondence: yangyi@cqut.edu.cn; Tel.: +86-139-0837-6853

**Abstract:** Aiming at the problems existing in the current radio energy transmission system, we propose a wireless power transmission (WPT) system with the parallel–parallel (PP)-compensated structure. The transmitter of the transmission system adopts a separate topological structure to suppress the current shock and noise. In order to improve the efficiency of the WPT, reduce the static loss, and reduce the current oscillation loss on the power side, the input current ripple can be improved by two parallel phase-shifting methods. In this paper, two topological theories are analyzed, and the simulation and experiment results verify the correctness of these theories under both static and on-load conditions. After the final two-way phase-shift, 61.99% of the ripple is reduced. It provides a new approach for the design of WPT systems with PP structure.

**Keywords:** parallel–parallel (PP)-compensated structure; isolating DC topological transmitter circuit; input current ripple; system efficiency





## 1. Introduction

In this paper, the wireless power transmission (WPT) system with the traditional parallel–parallel (PP)-compensated structure is improved. Since the inductance current and capacitor voltage cannot mutate, the compensation capacitor is applied with a constant voltage provided by the bias power supply when the circuit is turned on, but simultaneously the current flowing through the compensating capacitor is mutated [1–3]. Since the current flowing through the inductance cannot mutate, the direct current (DC) is blocked by the circuit and only the alternating current (AC) flows through the capacitor and inductance. Therefore, it has achieved the function of isolating DC (IDC) [4–6], and the circuit topology described above is called the IDC circuit topology. In this paper, we propose an IDC transmitter circuit for wireless charging which can effectively improve the input current waveform of the resonant circuit, reduce the damage of current oscillation to the devices, and improve the system efficiency [7–9].

## 2. System Structure and Operation Process

As shown in Figure 1, the system is composed of an automatic voltage rising and falling circuit, a current and voltage detection circuit, a high-frequency switching circuit, and a rectifier filter circuit and a DC–DC converter [10–14]. The micro controller unit (MCU) generates a PWM wave to control the on-off of the high-frequency switching tube so that the resonance between the inductor and capacitor at the transmitter can eliminate reactive power, and the active power can be coupled with the receiver as much as possible. The receiver inductance and capacitor resonant eliminates the reactive power losses so that the circuit efficiency reaches the highest and then provides power to the loads [15–17].

A schematic diagram of parallel resonance of an IDC transmitter is shown in Figure 2, where $C_1$ is the compensation capacitor, $L_1$ is the resonant inductance, and $V_{T1}$ is the switch

tube. The input DC voltage $V_{g1}$ is the stable DC voltage output after the input voltage of the total system passes through the voltage converter. For the convenience of analysis, all components in the circuit are assumed to be ideal components. Figure 3 is the topology of parallel resonance of the traditional transmitter, where $C_2$ is the compensation capacitor, $L_2$ is the resonant inductance, and $V_{T2}$ is the switching tube. The modal waveforms diagram of the transmitter circuit is shown in Figure 4. The input DC voltage $V_{g2}$ is the stable DC voltage output after the input voltage of the total system passes through the voltage converter. The modal waveforms diagram of the transmitter circuit is shown in Figure 5. For the convenience of analysis, all components in the circuit are assumed to be ideal components. The modal analysis of the two circuits is shown in Table 1.

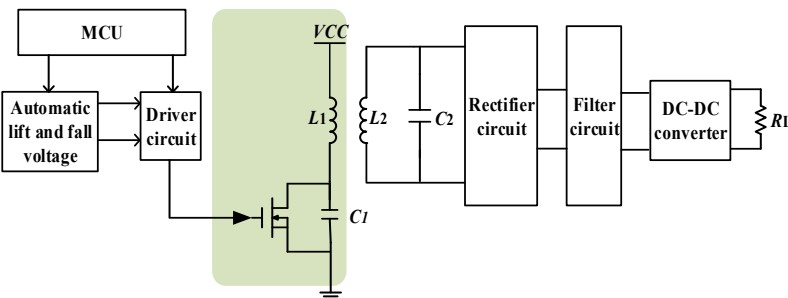

**Figure 1.** Block diagram of system structure.

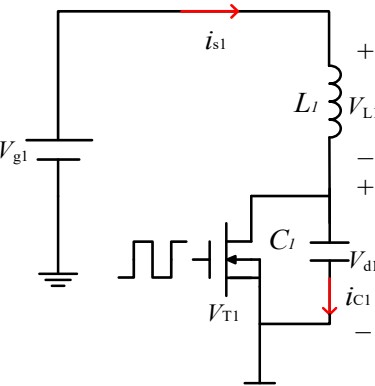

**Figure 2.** Schematic diagram of parallel resonance of an IDC transmitter circuit.

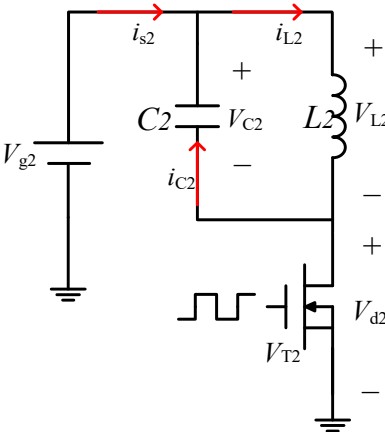

**Figure 3.** Schematic diagram of a traditional transmitter circuit.

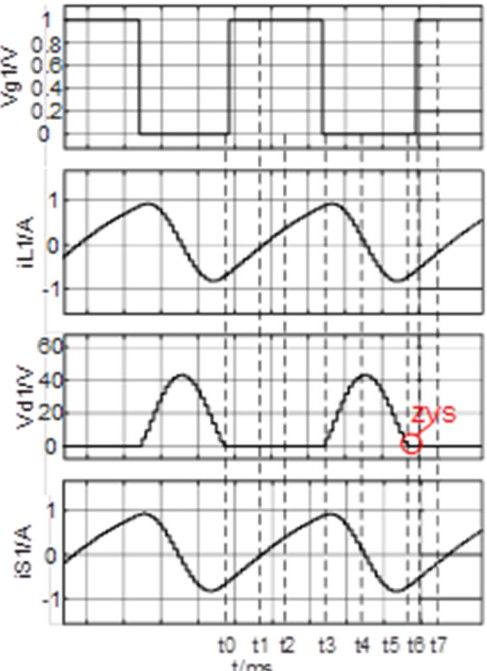

**Figure 4.** IDC transmitter modal waveforms.

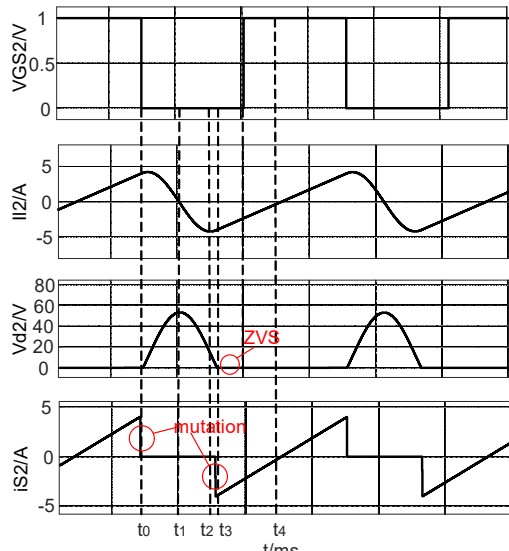

**Figure 5.** Traditional transmitter modal waveforms.

**Table 1.** Modal analysis of the transmitter.

| Mode | Separate Topological Transmitter | Traditional Topological Transmitter |
|---|---|---|
| Stage 1 | $i_{L1}$ increases linearly when the switch is on. | Switch tube continuation diode conduction. |
| Stage 2 | When the switch is off, $i_{L1}$ decreases and $C_1$ accumulates charge. | When the switch is on, $i_{L2}$ increases linearly. |
| Stage 3 | The switch is closed and the capacitor charges the inductance. | The switch tube closes and the inductance and capacitor enter the resonant state. |
| Stage 4 | The switch tube is turned on with zero voltage, and the inductance current is continued through the continuing diode, which is reduced to zero, and stage 1 is repeated. | The inductance charges the capacitor in reverse, and the voltage withstand of the switch reaches its maximum. |
| Stage 5 | | Capacitor discharges, voltage resistance of switch tube is reduced. |
| Stage 6 | | The inductance charges the capacitor, the voltage drop of the switch is zero, and the continuation secondary leads on. |
| Stage 7 | | The switch tube realizes zero voltage conduction. |

## 3. Circuit Model Analysis

DC and AC always exist in the amplifying circuits, and it is divided into DC path and AC path. In the DC path, the capacitor is opened and the inductance is shorted, and meanwhile AC signal source is also short [18]. In this case, the DC power supply of the capacitor in the AC path is shorted. Figure 6 shows the topological diagram of DC and AC paths of the IDC transmitter. In the AC path, the capacitance value is very small, thus it should not be regarded as a short circuit [19]. Figure 7 shows the micro-variant equivalent circuit diagram for the traditional transmitter. According to the above analysis method, it can be concluded that the DC and AC paths of the traditional transmitter are the same as those of the IDC circuit topology. Therefore, it can be deduced that the parallel resonant IDC circuit can be equivalent to the traditional parallel resonant circuit.

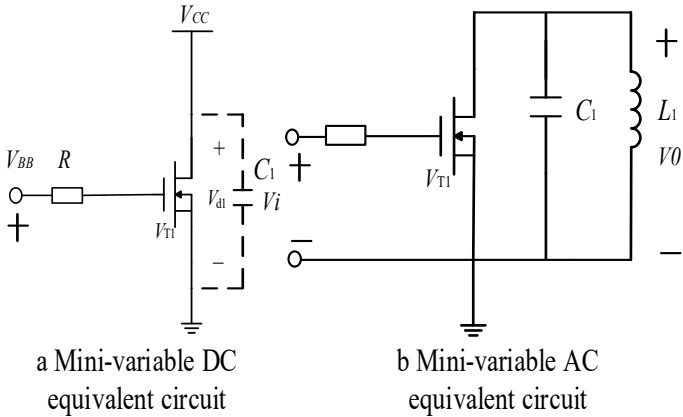

a Mini-variable DC
equivalent circuit

b Mini-variable AC
equivalent circuit

**Figure 6.** Micro-variant equivalent circuit diagram for IDC transmitter.

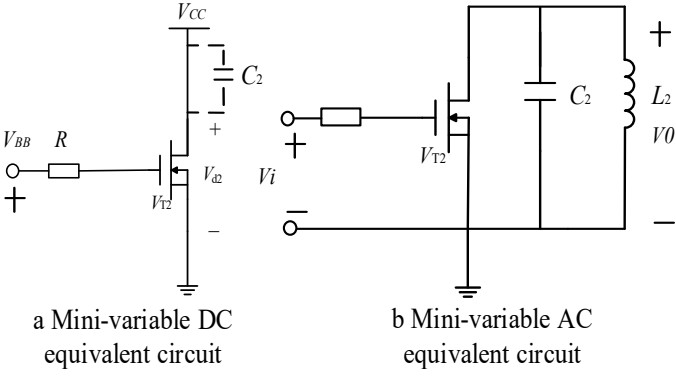

a Mini-variable DC
equivalent circuit

b Mini-variable AC
equivalent circuit

**Figure 7.** Micro-variant equivalent circuit diagram for the traditional transmitter.

Figure 8 shows the equivalent circuit model, where $L_P$ is the equivalent inductance, $R_P$ is the equivalent resistance, $U_{OC}$ is the open-circuit voltage of the side, $i_P$ is the inductance current, $L_S$ is the inductance of the side, $C_S$ is the compensating capacitor of the side, and R is the load, which equals to $R = \frac{\pi^2 R_L}{8}$. The secondary equivalent impedance is $Z_S = j\omega L_S + \frac{R}{1+j\omega C_S R}$ [20].

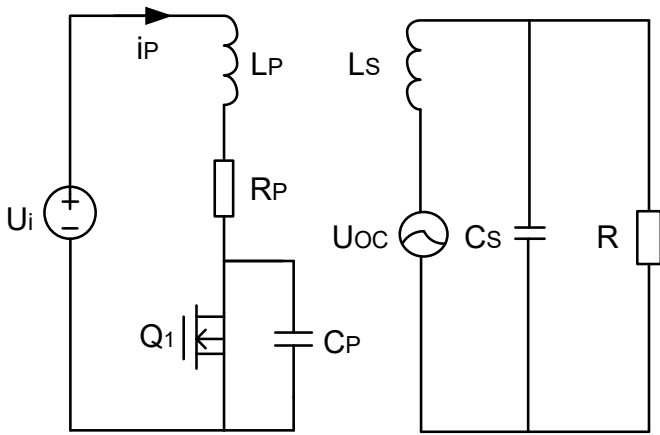

**Figure 8.** Equivalent circuit diagram of primary and secondary sides.

The equivalent reflection impedance of the secondary side to the primary side can be obtained as

$$Z_P = \frac{\omega^2 M^2}{Z_S} = jX_P + R_P \tag{1}$$

where $\omega$ is the angular frequency and M is the mutual inductance between the primary and secondary coils. From the above Figure 1, we have $L_P = L_1 + X_P/\omega$.

From Mode stage 1 described above, the switch conduction capacitance $C_P$ is shorted. Inductance $L_1$ is in charge state and the initial current is zero. According to Kirchhoff's voltage law, we have

$$i_{P(t)}R_P + L_P \frac{di_{P(t)}}{dt} = U_i \ t_0 < t < t_1 \tag{2}$$

The current through the inductance in Equation (2) $[t_0, t_1]$ is

$$i_{P(t)} = \frac{U_i}{R_P}\left[1 - e^{-\frac{R_P}{L_P}(t-t_0)}\right] \quad t_0 < t < t_1 \tag{3}$$

The peak current of inductance $L_1$ is

$$I_{PMAX} = \frac{U_i}{R_P}\left[1 - e^{-\frac{R_P}{L_P}DT}\right] \tag{4}$$

where $D$ is the duty ratio of tube and $T$ is the operating cycle.

When the switch tube is turned off, the inductance $L_1$ and capacitor $C_1$ will realize resonance. According to Kirchhoff's voltage law and current law, it can be known that

$$\begin{cases} C_P \frac{du_{CP(t)}}{dt} = i_{P(t)} \\ i_{P(t)}R_P + L_P \frac{di_{P(t)}}{dt} + u_{CP(t)} = 0 \end{cases} \quad t_1 < t < t_2 \tag{5}$$

Let $i_{P(t1)} = i_{PMAX}$, $u_{CP(t1)} = U_i$, the current and voltage of the inductance $L_1$ are formulated, respectively, as

$$\begin{cases} i_{P(t)} = -\sqrt{\frac{C_P}{L_P}} A e^{\alpha(t-t_1)} \sin[\beta(t-t_1) - \phi + \varphi] \\ u_{CP(t)} = A e^{\alpha(t-t_1)} \sin[\beta(t-t_1) - \varphi] \end{cases} \quad t_1 < t < t_2 \tag{6}$$

where

$A = \sqrt{U_i^2 + \left(\frac{I_{PMAX}}{\beta C_P} + \frac{\alpha U_i}{\beta}\right)^2}$,

$\alpha = -\frac{R_P}{2L_P}$,

$\beta = \sqrt{\frac{4L_P - R_P^2 C_P}{4L_P^2 C_P}}$,

$\phi = \arctan\left(\frac{U_i}{\frac{I_{PMAX}}{\beta C_P} + \frac{\alpha U_i}{\beta}}\right)$,

$\varphi = \arctan\left(\frac{\beta}{\alpha}\right)$.

According to Equation (6), the current $i_{CP}$ flowing through the capacitor $C_P$ is

$$i_{cp} = \begin{cases} 0 & t_0 < t < t_1 \\ C_P[-A \times \beta e^{\alpha(t-t_1)} \cos(\phi - \beta(t-t1)) + \\ A\alpha e^{\alpha(t-t_1)} \sin(\phi - \beta(t-t_1))] & t_1 \le t < t_2 \end{cases} \tag{7}$$

As shown in Figure 2, the input current $i_{S1}$ is equal to the current flowing through the inductance $L_1$ for the parallel resonant circuit of the IDC transmitter, such that

$$i_{s1} = \begin{cases} \frac{U_i}{R_P}\left[1 - e^{-\frac{R_P}{L_P}(t-t_0)}\right] & t_0 < t < t_1 \\ -\sqrt{\frac{C_P}{L_P}} A e^{\alpha(t-t_1)} \sin[\beta(t-t_1) - \phi + \varphi] & t_1 \le t < t_2 \end{cases} \tag{8}$$

According to Kirchhoff's current law (KCL), the total input current $i_{S2}$ equals to the sum of the current flowing through the capacitor $C_1$ and the inductance $L_1$

$$i_{s2} = \begin{cases} A e^{\alpha(t-t_1)} \begin{bmatrix} -\beta \cos(\phi - \beta(t-t_1))C_P + \alpha \sin(\phi - \beta(t-t_1))C_P + \\ \sin(\phi - \varphi - \beta(t-t_1))\sqrt{\frac{C_P}{L_P}} \end{bmatrix} & t_0 < t < t_1 \\ 0 & t_1 \le t < t_2 \end{cases} \tag{9}$$

## 4. The Simulation Analysis

### 4.1. Simulation Analysis of Input Current

In order to verify the effectiveness and feasibility of the proposed IDC transmitter circuit, the Matlab platform is used to simulate and analyze the traditional PP wireless charging circuit and the IDC PP wireless charging circuit [21,22]. According to the working principle of the PP wireless charging system, the main circuit parameters of the system are calculated, as shown in Table 2.

**Table 2.** Main parameters of the simulation.

| Parameter Name | Parameter Value |
|---|---|
| Switching frequency | 206 kHz |
| Receiving/Transmitting resonant inductance | 7 μH |
| The receiver compensates the capacitance | 0.094 μF |
| load | 15 Ω |
| The coupling coefficient | 0.3 |

This model mainly analyzes the waveform changes of each device in the transmitter circuit. The output voltage of the automatic voltage rising and falling circuit at the input end is replaced by the DC voltage source that has the steady voltage output, and the corresponding input voltage is set to 12 V and the compensation capacitor in the traditional parallel resonant transmitter circuit is set to 0.094 μF. In the IDC circuit, the filter capacitor and compensation capacitor are both set to 0.047 μF in the transmitter side.

Figure 9 compares the inductance current between the parallel resonant circuits for the IDC transmitter and the traditional transmitter. It can be seen that the waveforms of $i_{L1}$ and $i_{L2}$ are the same, and the conclusion drawn above can be achieved that parallel resonant circuits for the IDC transmitter are equivalent to resonant circuits for the traditional transmitter through this simulation.

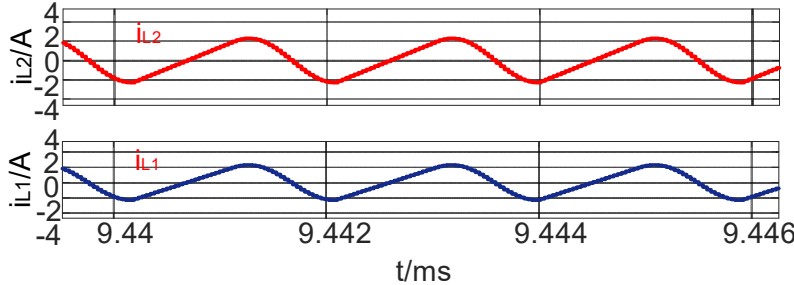

**Figure 9.** Inductive current waveform for the two topologies.

Figures 10 and 11 show the input current waveform of the traditional and IDC parallel resonance circuits in static mode, respectively. In the figures, $i_{S2n}$ is the input current waveform in practice for the traditional circuit. $i_{S1n}$ is the input current waveform in practice for the IDC circuit. $i_{S2'n}$ is the input current waveform under ideal conditions for the traditional circuit. The reason for the difference between these two waveforms is that there is a deviation between the theoretical situation and the actual situation. In theoretical analysis, the switch tube is often considered an ideal element, and the parasitic capacitance in the switch tube is ignored, resulting in the $i_{S2'n}$ waveform. However, in practice, there is a parasitic capacitor inside the open light tube, and when the circuit is in the resonant state, the parasitic capacitor will release power to the resonant circuit, thus mitigating the degree of the sudden change of the input current and smoothing the current waveform. This discovery is proposed for the first time in this paper, and it is hoped that it will be used as a reference for future research and analysis.

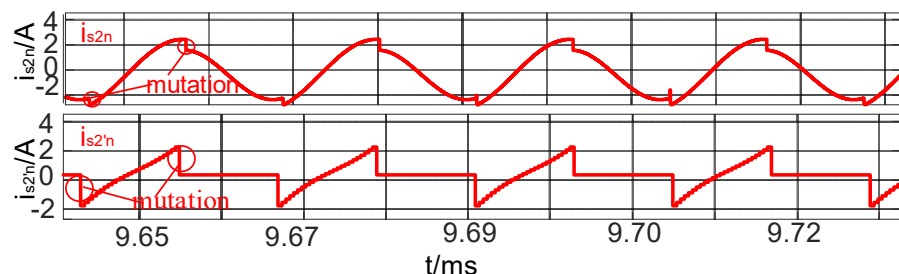

**Figure 10.** The input current waveform of traditional parallel resonance circuit in static mode.

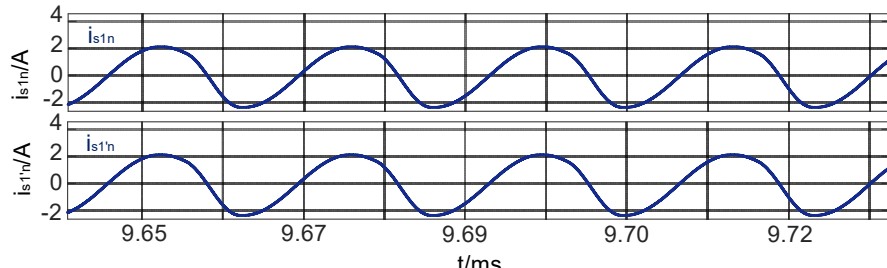

**Figure 11.** The input current waveform of the IDC parallel resonance circuit in static mode.

The two groups of simulation waveforms $i_{S1n}$ and $i_{S2n}$ in the figure are the same as those derived from the theoretical model, and the input current waveforms of the resonant unit are significantly different. It can be seen from the figure that there is no difference in the input circuit waveforms of the two kinds of circuits in the time period $[t_0, t_1]$. This is because the capacitor current of the switch tube is shorted at this time, so the capacitor current is zero. For the traditional parallel resonance, the input current $i_{S2n}$ in the time period $(t_0, t_1)$ is equal to the inductance current $i_{L2}$. When the switch tube is opened in the time period $(t_1, t_2)$, the resonant input current generated by the power exchange between capacitor and inductance is equal to zero. However, for the IDC parallel resonant circuit, the input current $i_{S1n}$ is always equal to the inductive current $i_{L1}$, so that there will be no sudden change, and the current waveform is smoother. There is no current oscillation brought by the original circuit, and the current oscillation will cause great damage to the power supply. The IDC circuit protects the power supply and improves the charging efficiency.

Figure 12 is the input current waveform of the traditional in parallel resonance circuit on load mode, and Figure 13 is the input current waveform of the IDC in parallel resonance circuit on load mode. As can be seen from the figure, considering the losses of devices and loads, the simulated waveform is basically consistent with the theoretical waveform. It can be seen from the simulation waveform that the input current waveform of the power supply side has a mutation after the traditional parallel resonant circuit is on load, and there is a large amount of charge that cannot be released in the circuit, which leads to a great current conflict. The input current waveform of the IDC is the same as the resonant inductance waveform, which effectively reduces the loss on the power source side during LC oscillation, thus protecting the devices on the power side. Moreover, the waveform can be measured, which plays an important role in the closed-loop regulation part of the system. Figure 12 shows that the IDC has a stronger load-carrying capacity than the traditional topology.

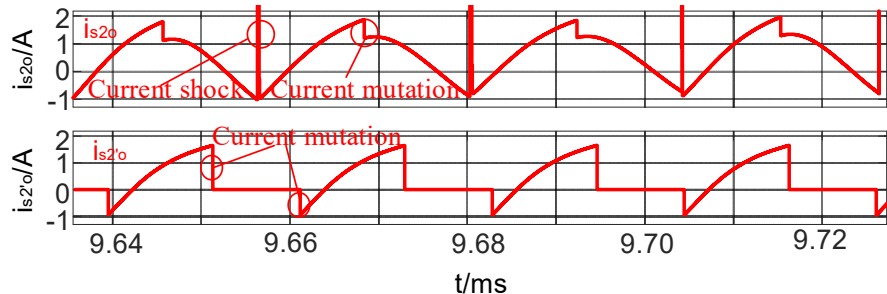

**Figure 12.** The input-current waveform of traditional in parallel resonance circuit on load mode.

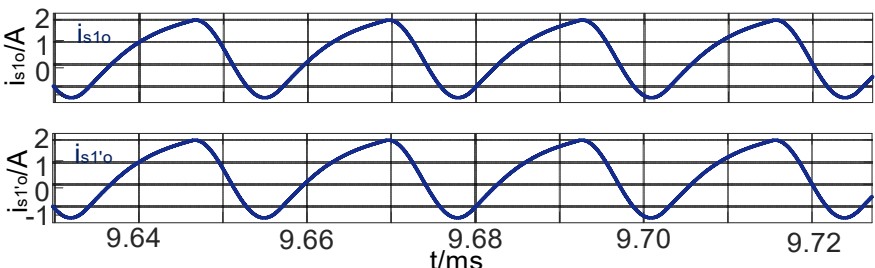

**Figure 13.** The input-current waveform of IDC parallel resonance on load mode.

*4.2. Simulation Analysis of Ripple Suppression*

Due to AC/DC mixing phenomenon in the amplifier circuit and the existence of inductance and capacitance elements in the resonant circuit, so when the switch is on, the circuit charges the inductance $L_1$ and current flows into the inductance. When the switch tube is cut off, the inductance $L_1$ and the compensation capacitor $C_1$ begin to resonate. When the capacitor $C_1$ transfers all the power to the inductance $L_1$, the current will flow backward into $L_1$. Since the input current $i_{S1}$ is equal to the inductance current $i_{L1}$, the input current will increase in the opposite direction, which will cause the ripple in the circuit.

Ripple has a serious impact on the stability of the whole system and will cause great damage to components and directly affects the efficiency of the system by causing disadvantages such as heating of the whole system. Therefore, research on ripple suppression is especially necessary. If the ripple can be reduced, it will make a great contribution to the improvement of efficiency. In this paper, based on Kirchhoff's current law, the topology of the transmitter in parallel and output in series is adopted. As shown in Figure 14, taking parallel phase-shifting of two transmitting terminals as an example, the two modules have the same parameters, and the specific parameters are shown in Table 1.

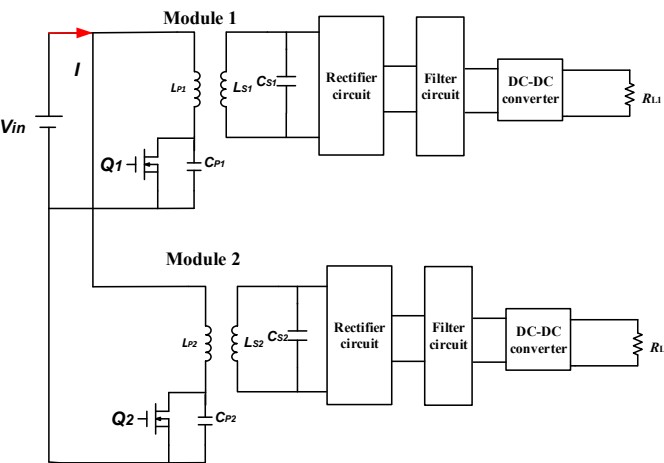

**Figure 14.** Two-way phase-shift topology.

Figure 15a shows the relationship among a phase-shifting angle, switching frequency and current peak. The phase-shifting angle is in the range of 10–350 degrees, and the switching frequency is in 180–230 K. It can be seen from the figure that when the phase-shifting angle is in the range of 170–190 degrees and the switching frequency is in the range of 196–215 K, the current peak is in the trough state. In addition, with the increase in phase-shifting angle and frequency, the current peak value has an obvious trend of increase. In order to further analyze the relationship between the three optimal points of phase-shifting, frequency was found and a three-dimensional scatter diagram was made as shown in Figure 15b. In this figure, the phase-shifting angle was in the range of 170–190 degrees and the frequency was at 196–215 K. It can be seen from the figure that the current peak value is the smallest when the phase-shifting angle is 180 degrees and the frequency is 206 K.

In addition, before 180 degrees, the smaller the angle, the bigger the peak, and after 180 degrees, the bigger the angle, the bigger the peak.

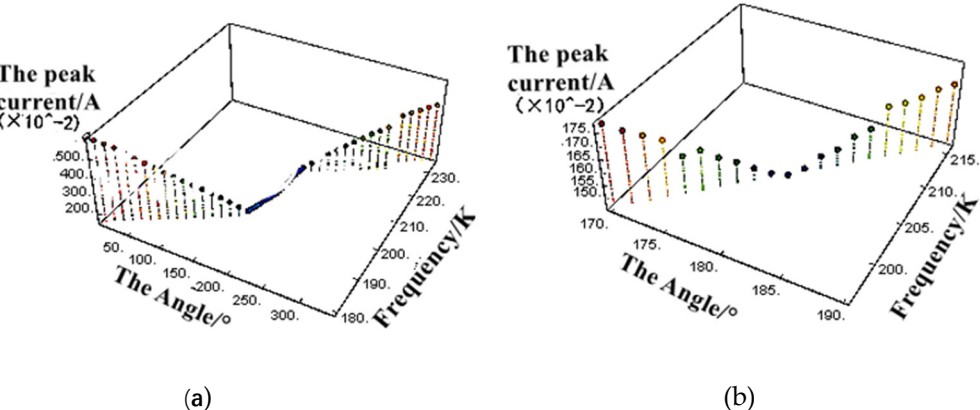

(a)　　　　　　　　　　　　　　　　　　(b)

**Figure 15.** Relationship between circulation size, phase-shifting angle, and switching frequency. (**a**) Anlarged; (**b**) Zoom out.

　　　Figure 16 shows the input current waveform after phase-shift at 180 degrees in the static condition of the traditional topology. As shown in the figure, $i_{S2n}$ is the input current waveform before phase-shift and $i_{S2'n}$ is the input current waveform after phase-shift. Through comparison, it can be seen that the input current amplitude of the traditional topology has not been improved after the phase-shift, and the waveform has a mutation and distortion. Its fundamental properties have not been changed, and it seems to be more fluctuant without any help to the stability of the whole system. Figure 17 shows the input current waveform of the IDC at 180 degrees phase-shift under static condition. In the figure, $i_{S1n}$ is the input current waveform before the phase-shift of the IDC, and $i_{S1'n}$ is the input current waveform after the phase-shift of the IDC. It can be seen from the figure that the peak value of the input current of the parallel resonant circuit of the IDC decreases significantly after the phase-shift of 180 degrees, and the current is similar to the sine wave, which is very smooth. This is of great help to the stability of the whole system, which can greatly improve the system's efficiency and reduce the loss of power supply.

　　　As shown in Figure 18, $i_{S2O}$ is the input current waveform before phase-shifting in the traditional topology, and $i_{S2'o}$ is the input current waveform after phaseshifting in the traditional topology. It can be seen that the amplitude does not change much after phase-shifting in the traditional topology, and the mutation and conflict of the current become more serious. As shown in Figure 19, $i_{S1o}$ is the input current waveform before phase-shifting in the IDC topology, and $i_{S1'o}$ is the input current waveform after phase-shifting in the IDC topology. It can be seen that the current amplitude drops significantly, and the waveform becomes smooth after phase-shifting in the IDC topology.

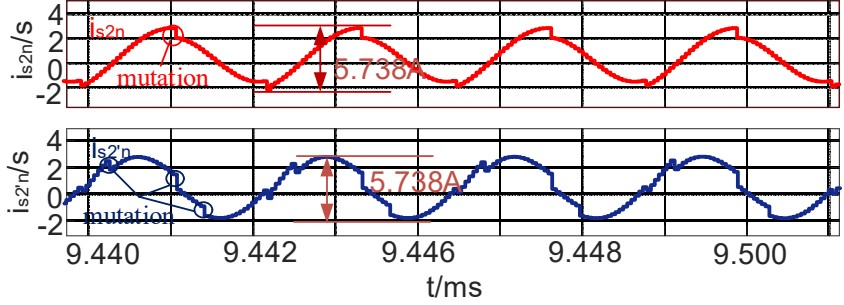

**Figure 16.** Traditional topological static 180° phase-shifting input current waveform.

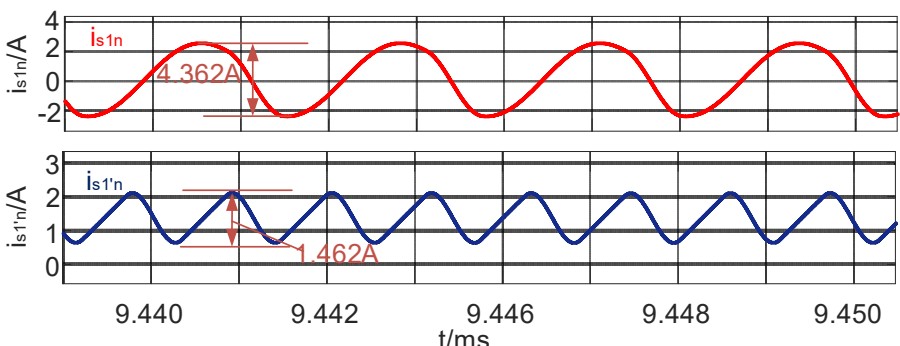

**Figure 17.** IDC static 180° phase-shifting input current waveform.

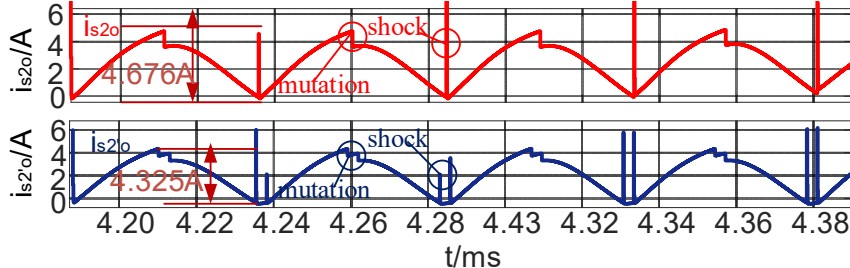

**Figure 18.** Traditional topological on-load 180° phase-shifting input current waveform.

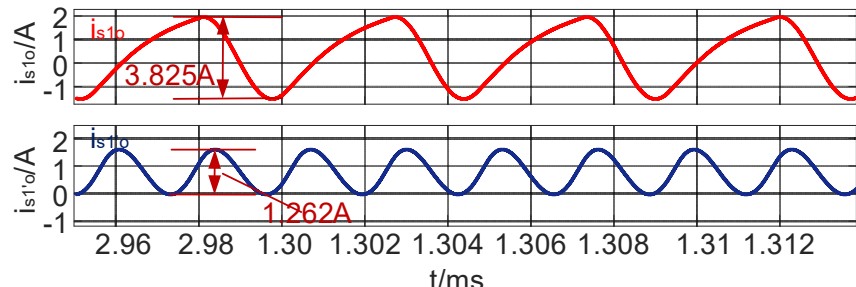

**Figure 19.** IDC on-load 180° phase-shifting input current waveform.

## 5. Experimental Verification

In this paper, experimental devices for wireless power transmission are applied for experimental verification. Figure 20 shows the physical diagram of the IDC wireless power transmission system. The input DC voltage is supplied by an external power supply, the inductance value is 7 µH, the compensation capacitance of the receiver is 47 nF, and the working frequency of the fixed system is 206 KHz in the experimental test.

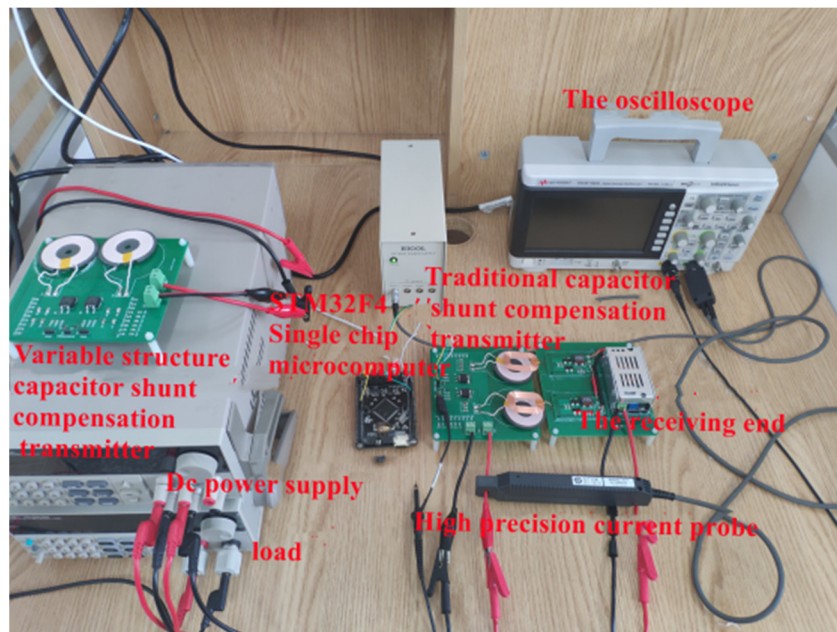

**Figure 20.** Physical diagram of hardware.

The traditional parallel resonant circuit and the IDC wireless charging circuit were constructed for experimental verification. Figures 21 and 22 present the two topological soft switching waveforms. Both the two topological circuits can realize zero-voltage conduction, and the drive waveform and the voltage stress waveform of the switch tube of IDC are smoother. As shown in Figures 23 and 24, the single-channel input current waveform of the two groups of circuits under no-load condition is measured. As shown in Figures 25 and 26, the input current waveform under two-channel phase-shifting condition is measured. As shown in Figures 27 and 28, the inductance current waveform under no-load is measured. As can be seen from the figure, the distortion of input current waveform for the traditional topology is serious and the current peak value does not change after phase-shift. The single-phase input current of the IDC is smoother, which is similar to the sine wave, and the current ripple is effectively reduced after the two-way phase-shifting. The simulation and physical test demonstrate that the peak value of the input current of the traditional topological circuit is higher than that of the IDC, which is caused by the combined action of the compensating capacitor and the parasitic capacitor of the switch tube.

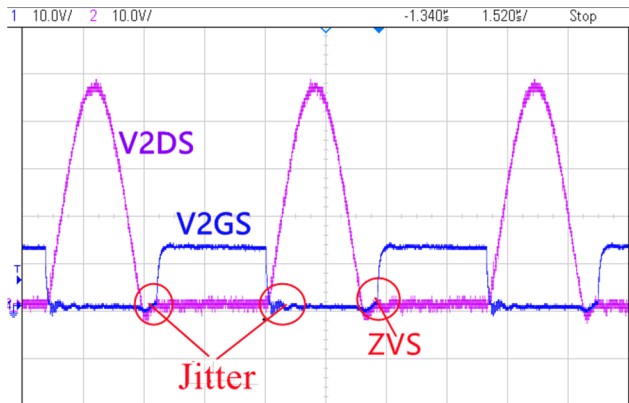

**Figure 21.** Traditional topological soft switch waveforms.

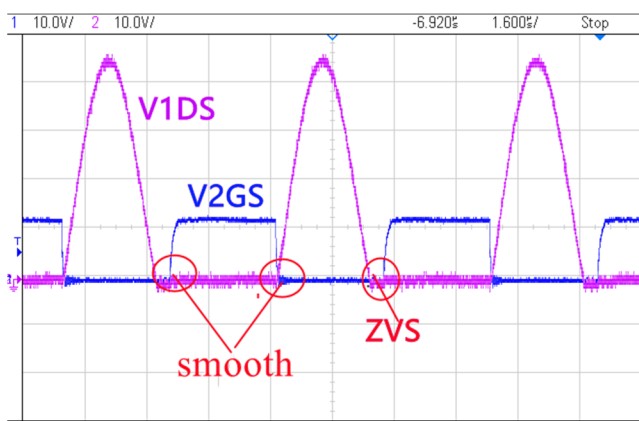

**Figure 22.** IDC soft switch waveforms.

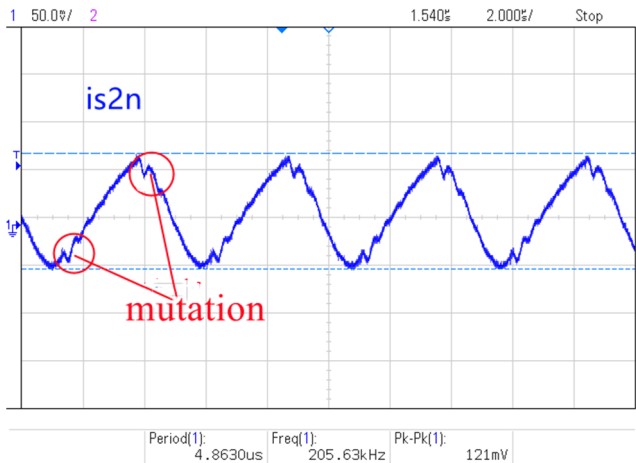

**Figure 23.** Traditional topology static.

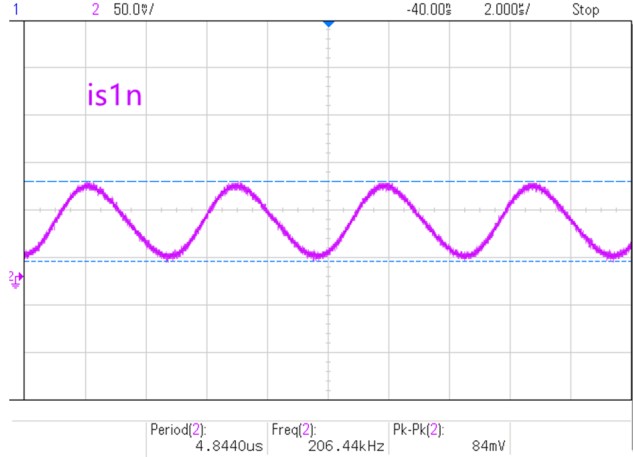

**Figure 24.** IDC static input current waveform.

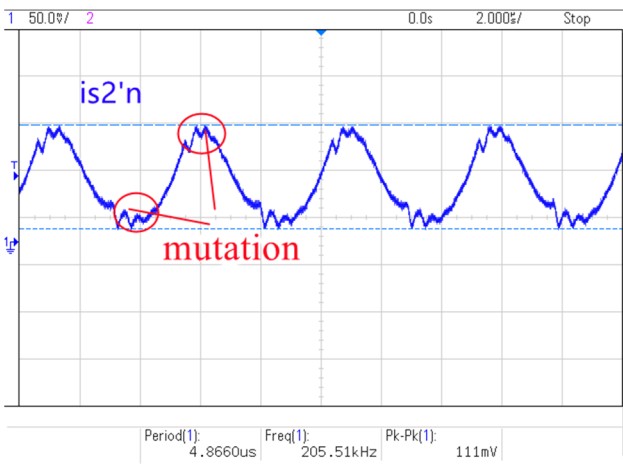

**Figure 25.** Traditional topological static 180° shifting input current waveform.

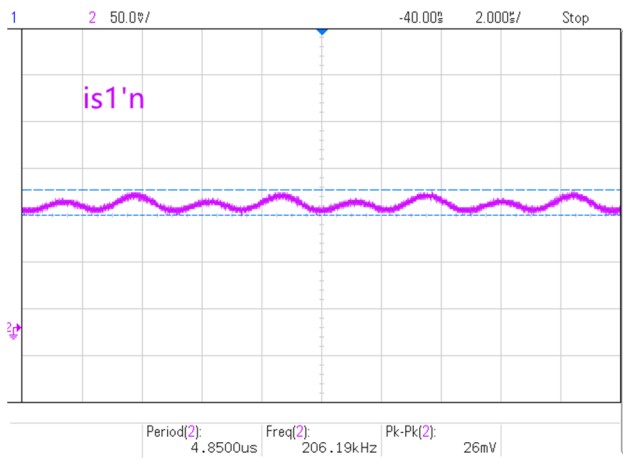

**Figure 26.** IDC static 180° phase-shifting input phase-current waveform.

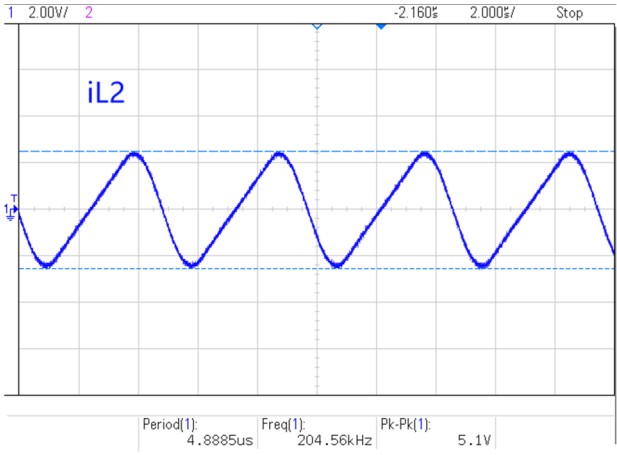

**Figure 27.** Traditional topological inductive current waveform.

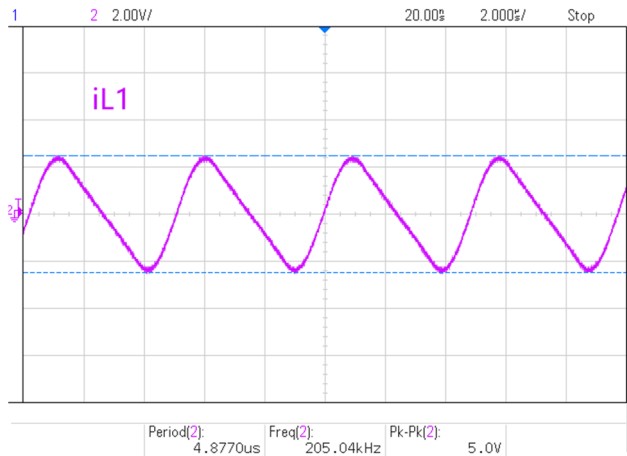

**Figure 28.** IDC inductive current waveform.

Figure 29 shows the single-phase input current waveform with load for the traditional topology, Figure 30 is the single-phase input current waveform with load for IDC topology, Figure 31 is the two-way phase-shifting input current waveform with load for the traditional topology, and Figure 32 is the two-way input current waveform with load for IDC topology. Comparing Figures 24 and 30, we can find that there is no obvious distortion in the waveform of the IDC with load. According to Figures 23 and 29, the input current waveform is obviously distorted after the traditional topology is on load, indicating that the IDC has a stronger load-carrying capacity. It can be seen from Figures 26 and 32 that the waveforms are still smooth without distortion after two-way phase-shifting with load in the IDC. According to Figures 25 and 31, the waveforms mutate more seriously after two-way phase-shifting with load in the traditional topology. According to Figures 29–32, the waveform of the IDC is still smooth after the load is accessed, but the input current waveform of the traditional topology has obvious fluctuation. Under on-load condition, the input current amplitude of traditional topological circuit is larger than that of IDC.

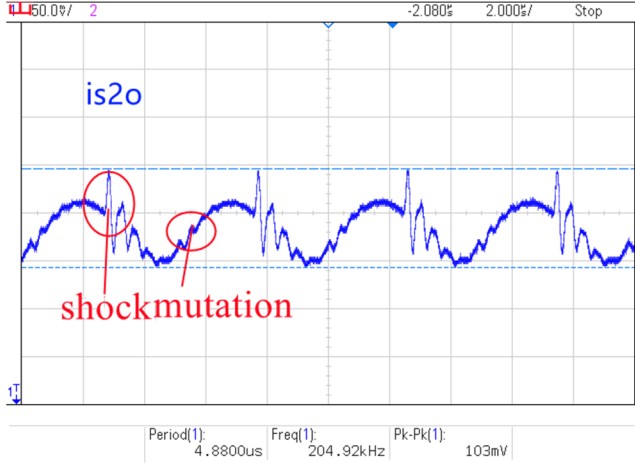

**Figure 29.** Input current waveform under of traditional topology.

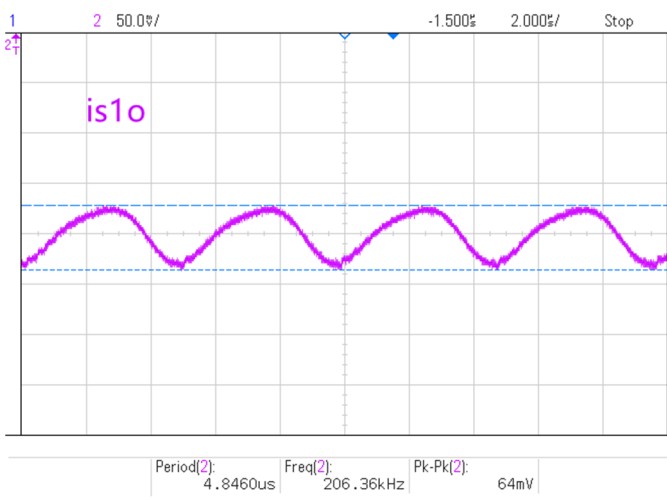

**Figure 30.** Input current wave form under load condition condition of IDC.

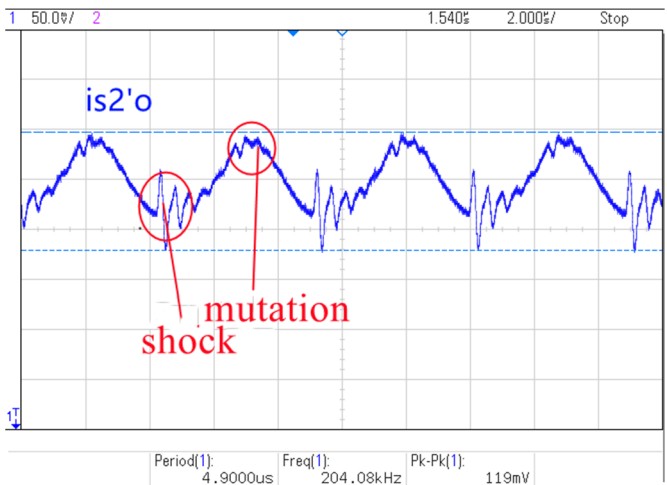

**Figure 31.** Waveform of 180° phase-shifting input current under load condition of traditional topology.

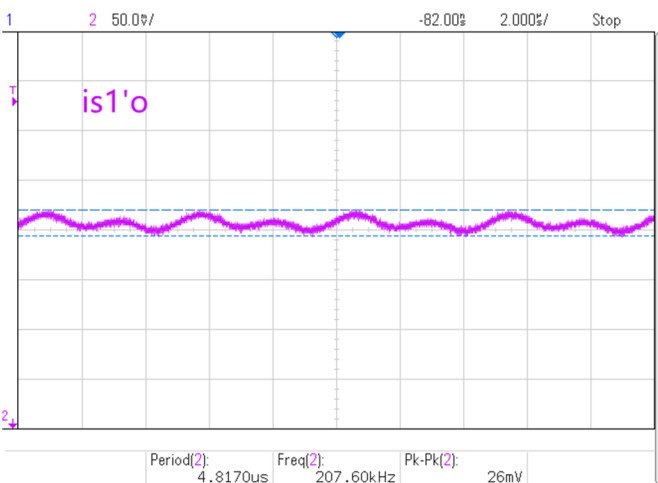

**Figure 32.** Waveform of 180° phase-shifting input current under load condition of IDC.

The transmission efficiency of the two topologies was measured under the condition of input voltage of 12 V and output voltage of 5 V. The results are shown in Table 3. The experimental data clearly show that the transmission efficiency of the system can be

improved by using the IDC transmitter circuit, which verifies the feasibility of the proposed variable topology transmitter circuit.

**Table 3.** Transmission efficiency data measurement.

|  | Input Current/A | Output Voltage/V | Output Current/A | Transmission Efficiency/% |
|---|---|---|---|---|
| Traditional PP system | 1.05 | 5.25 | 2.01 | 83.75 |
| IDC PP-type system | 1.03 | 5.21 | 2.02 | 85.15 |

## 6. Conclusions

In this paper, a PP resonant wireless power transmission system is proposed, and the input current model of input current, inductance current, and capacitance voltage are obtained from the two topologies by modal analysis and mathematical modeling of the transmitter topology. The correctness of the mathematical model of IDC topology proposed in this paper is verified by simulation results. In addition, the experimental current waveform also reflects that the IDC circuit has a significant improvement effect on the input current waveform of the system. In order to understand the ability of the two topologies to suppress the ripple, Matlab/Simulink simulation shows that the IDC topology with the two phases shifting at 180 degrees has a stronger ability for ripple suppression. Finally, a traditional parallel resonant wireless charging system and IDC wireless charging system platform are built. Setting the operating frequency to 206 kHz and the input voltage to 12 V, the experimental waveforms of resonant inductance current, switch voltage, and input power supply of the two groups of circuits are obtained under no-load and on-load conditions. It is verified that the IDC can effectively reduce the static losses of the system, improve the charging efficiency, and suppress the input current ripple, which plays a better role in the research of the closed-loop control strategy of the system.

**Author Contributions:** Conceptualization, Y.Y., X.Z. and L.L; formal analysis, S.X. and Q.Z.; investigation, X.Z.; methodology, L.L. All authors have read and agreed to the published version of the manuscript.

**Funding:** This research was funded by Chongqing technical innovation and application development special project (cstc2019jscx-msxmX0003) and Science and Technology Research Project of Chongqing Education Commission (KJQN201801142).

**Data Availability Statement:** The datasets used and/or analyzed during the current study are available from the corresponding author on request.

**Acknowledgments:** We want to thank national active distribution network technology research center of Chongqing University of Technology for the support in the completion of the project.

**Conflicts of Interest:** The authors declare no conflict of interest.

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
