# Peer review of "Research on High Power Factor Single Tube Variable Structure Wireless Power Transmission"

_wevj, doi:10.3390/wevj12040214_

Round 1

Reviewer 1 Report

This research work is quite interesting. However, the submitted manuscript lacks many of the essential components of a sound academic paper.

Some comment(s) or suggestion(s):

1)

Lines 13 to 14 in the abstract need to be reworded for clarity. What following methods?

2)

The abstract needs to capture, summarily, the findings (including the metrics verifying the improvement offered by the authors' method in comparison to existing methods) in the paper.

3)

The introduction needs to provide a sufficient background into the authors' investigations: the rationale for the work carried out, the research gap identified in the research area, a critical review of existing similar works in the research area, and the main contributions of the paper highlighted summarily.

4)

In Section 2, the implication of the assumption made in Lines 53 to 54 needs to be discussed in the context of real-world applications to justify the practicality of the work from this viewpoint.

5)

In Section 3, what analysis method is being referred to in Line 83? More analytical details should be provided to support the deductions made about the circuit model by the authors.

6)

Equations 1 to 9 and their associated schematics need to be properly linked to the opening paragraph of Section 3 to ensure a good and proper flow. All parameters and variables in all these equations should be clearly defined and explained.

7)

In Section 4, the simulation set up in MATLAB should be discussed prior to the simulation analysis.

8)

There are no calculations in Table 1, so Lines 136 to 137 in Section 4 could be wrong.

9)

In Section 5, the experimental setup should be discussed prior to the experimental verifications.

10)

The manuscript should be thoroughly proofread to address grammatical and typographical errors. E.g., Line 160 on Page 7, there is a capacitor is inside the open light tube, Lines 281 and 282 on Page 11, Figures 25 and 26 (not Figure 25 and 26), Figures 27 and 28 (not Figure 27 and 28)

11)

Authors need to be consistent. Fig. X or Figure X? Always leave a gap before stating the figure number.

12)

Leave a gap between numerical values and their units. Particularly, for frequency values. E.g., Table 2, 268 kHz.

Author Response

Respose to the Review Comments

We would like to thank you for your careful reading, helpful comments, and constructive suggestions, which has significantly improved the presentation of our manuscript.

We have carefully considered all comments from the reviewers and revised our manuscript accordingly. The manuscript has also been double-checked, and the typos and grammar errors we found have been corrected. In the following section, we summarize our responses to each comment from the reviewers. We believe that our responses have well addressed all concerns from the reviewers. We hope our revised manuscript can be accepted for publication.

This research work is quite interesting. However, the submitted manuscript lacks many of the essential components of a sound academic paper.

Response:

We thank the reviewer for reading our paper carefully and giving the above positive comments.

Case1

Lines 13 to 14 in the abstract need to be reworded for clarity. What following methods?

Response:
Thank you for pointing out this problem in manuscript. We have changed the original sentence to“In order to improve the efficiency of WPT, reduce the static loss, and reduce the current oscillation loss on the power side, the input current ripple can be improved by two parallel phase-shifting methods.”

Case 2

The abstract needs to capture, summarily, the findings (including the metrics verifying the improvement offered by the authors' method in comparison to existing methods) in the paper.

Response:

Thank you for pointing out this problem in manuscript. We have summarized the findings in the paper in the abstract.

Case 3

The introduction needs to provide a sufficient background into the authors' investigations: the rationale for the work carried out, the research gap identified in the research area, a critical review of existing similar works in the research area, and the main contributions of the paper highlighted summarily.

Response:

Thank you for pointing out this problem in manuscript. We have made changes to the introduction.

Case 4

In Section 2, the implication of the assumption made in Lines 53 to 54 needs to be discussed in the context of real-world applications to justify the practicality of the work from this viewpoint.

Response:

Thank you for pointing out this problem in manuscript. We have already made an addition to the second section.

Case 5

In Section 3, what analysis method is being referred to in Line 83? More analytical details should be provided to support the deductions made about the circuit model by the authors.

Response:

Thank you for pointing out this problem in manuscript. What we quoted is the small signal model analysis method, which has been explained in the article.

Case 6

Equations 1 to 9 and their associated schematics need to be properly linked to the opening paragraph of Section 3 to ensure a good and proper flow. All parameters and variables in all these equations should be clearly defined and explained.

Response:

Thank you for pointing out this problem in manuscript. We have made the corresponding explanation in the article.

Case 7

In Section 4, the simulation set up in MATLAB should be discussed prior to the simulation analysis.

Response:

Thank you for pointing out this problem in manuscript. We have made the corresponding explanation in the article.

Case 8

There are no calculations in Table 1, so Lines 136 to 137 in Section 4 could be wrong.

Response:

Thank you for pointing out this problem in manuscript. These data are obtained through formula derivation, modeling simulation and experimental verification, so they are correct.

Case 9

In Section 5, the experimental setup should be discussed prior to the experimental verifications.

Response:

Thank you for pointing out this problem in manuscript. We have already set up an experiment.

Case 10

The manuscript should be thoroughly proofread to address grammatical and typographical errors. E.g., Line 160 on Page 7, there is a capacitor is inside the open light tube, Lines 281 and 282 on Page 11, Figures 25 and 26 (not Figure 25 and 26), Figures 27 and 28 (not Figure 27 and 28).

Response:

Thank you for pointing out this problem in manuscript. We have checked and revised the English grammar as required.

Case 11

Authors need to be consistent. Fig. X or Figure X? Always leave a gap before stating the figure number.

Response:

Thank you for pointing out this problem in manuscript. We have made changes as required.

Case 12

Leave a gap between numerical values and their units. Particularly, for frequency values. E.g., Table 2, 268 kHz.

Response:

Thank you for pointing out this problem in manuscript. We have made changes as required.

Reviewer 2 Report

In this paper a new topology for the transmitter side of a WPT system is presented, analyzed and compared to a more traditional topology.

First of all, the "new topology" does not seem very new to me. So there is little scientific impact in this paper.

The simulation and verification seem to be correct, but the presentation is not adequate for a scientific paper. In many graphs and also in the text, indices are not written as indices. The signal graphs seem to be exported directly from the simulation tool or oscilloscope and the lettering and grid look awkward.

Therefore I cannot recommend this paper for publication

Author Response

Respose to the Review Comments

We would like to thank you for your careful reading, helpful comments, and constructive suggestions, which has significantly improved the presentation of our manuscript.

We have carefully considered all comments from the reviewers and revised our manuscript accordingly. The manuscript has also been double-checked, and the typos and grammar errors we found have been corrected. In the following section, we summarize our responses to each comment from the reviewers. We believe that our responses have well addressed all concerns from the reviewers. We hope our revised manuscript can be accepted for publication.

First of all, the "new topology" does not seem very new to me. So there is little scientific impact in this paper.

The simulation and verification seem to be correct, but the presentation is not adequate for a scientific paper. In many graphs and also in the text, indices are not written as indices. The signal graphs seem to be exported directly from the simulation tool or oscilloscope and the lettering and grid look awkward.

Therefore I cannot recommend this paper for publication.

Response:
Thank you for pointing out this problem in manuscript. IDC topology is to use small signal analysis method to avoid the sudden change of current caused by traditional topology, and the parallel capacitor reduces the switching stress, which has a good protective effect on the power supply and the switching tube. Our team will devote more energy to research topology in the future, and your suggestions are of great help to us. We will continue to find scientifically meaningful research content, thank you for your suggestions.

Reviewer 3 Report

This paper mainly studied a two-way phase shift topology for wireless power transfer with reduced ripples.  The content and results are both solid. Some minor concerns are listed as follows.

  1. The survey of relevant literature is not sufficient. Please add more discussion in Introduction and enrich the Reference.
  2. How about the voltage stress of switch in the used topology? Please discuss more.
  3. The operating frequency is 206 kHz. What are the criteria to use such a frequency?
  4. In Fig. 14, please discuss the possibility of implementing the multifrequency wireless power transfer.
  5. In Table 3, the transmission efficiency is not high, why?

Author Response

Respose to the Review Comments

We would like to thank you for your careful reading, helpful comments, and constructive suggestions, which has significantly improved the presentation of our manuscript.

We have carefully considered all comments from the reviewers and revised our manuscript accordingly. The manuscript has also been double-checked, and the typos and grammar errors we found have been corrected. In the following section, we summarize our responses to each comment from the reviewers. We believe that our responses have well addressed all concerns from the reviewers. We hope our revised manuscript can be accepted for publication.

This paper mainly studied a two-way phase shift topology for wireless power transfer with reduced ripples.  The content and results are both solid. Some minor concerns are listed as follows.

Response:

We thank the reviewer for reading our paper carefully and giving the above positive comments.

Case1

The survey of relevant literature is not sufficient. Please add more discussion in Introduction and enrich the Reference.

Response:

Thank you for pointing out this problem in manuscript. We have enriched the citations of references.

Case2

How about the voltage stress of switch in the used topology? Please discuss more.

Response:

Thank you for pointing out this problem in manuscript. In the traditional topology, due to the parallel connection of inductance and capacitance. Parallel resonance will cause a sudden change in the current, and the stress on both ends of the switch tube is very large and cannot be released. In the IDC topology, the capacitor is connected to both ends of the switch tube in parallel, so the stress of the switch tube is absorbed and the stress is reduced.

Case3

The operating frequency is 206 kHz. What are the criteria to use such a frequency?

Response:

Thank you for pointing out this problem in manuscript. The basis of this standard is the result of repeated demonstrations in our experiments.

Case4

In Fig. 14, please discuss the possibility of implementing the multifrequency wireless power transfer.

Response:

Thank you for pointing out this problem in manuscript. What we are currently studying is the result under balanced load. If the frequency is different when the load is different, multi-frequency wireless power transmission is completely achievable.

Case5

In Table 3, the transmission efficiency is not high, why?

Thank you for pointing out this problem in manuscript. Because we did not use ferrite in the coil during the research, and our circuit design did not meet the industrial design standard, the energy loss was too large.

Round 2

Reviewer 1 Report

The authors should consider providing a discussion on the new references [17 - 22]. It is not enough to just add them to the list of references. They should be discussed too. This will make the paper self-contained and much more sound.

Author Response

Respose to the Review Comments

We would like to thank you for your careful reading, helpful comments, and constructive suggestions, which has significantly improved the presentation of our manuscript.

We have carefully considered all comments from the reviewers and revised our manuscript accordingly. The manuscript has also been double-checked, and the typos and grammar errors we found have been corrected. In the following section, we summarize our responses to each comment from the reviewers. We believe that our responses have well addressed all concerns from the reviewers. We hope our revised manuscript can be accepted for publication.

This research work is quite interesting. However, the submitted manuscript lacks many of the essential components of a sound academic paper.

Response:

We thank the reviewer for reading our paper carefully and giving the above positive comments.

Case1

The authors should consider providing a discussion on the new references [17 - 22]. It is not enough to just add them to the list of references. They should be discussed too. This will make the paper self-contained and much more sound.

Response:

Thank you for pointing this out in the manuscript. We added references [17-22] to the corresponding positions in the article, and made explanations and marks.

Reviewer 2 Report

The form of the manuscript has to be improved.

I cannot go through all the details, just a few examples:

page 6, line 144: first µF is used, then uF

Figure9: space missing. In the figure, the numbers are overlapping and the quality is bad. IL1 and IL2 must be written as indices

Indices are not used in the complete text and the graphs. On the other hand, indices are mostly correct in the equations. This makes the manuscript hard to read.

Also many equations look strange, e.g. Eq. (8): there is no "=" in this equation. Also brackets are not set in a proper way. The [ ] is too small in Eq. (8), just as an example. If the manuscript is written in latex, you should use \left[ instead of [.

These things have to be corrected befor the manuscript is readable.

Author Response

Respose to the Review Comments

We would like to thank you for your careful reading, helpful comments, and constructive suggestions, which has significantly improved the presentation of our manuscript.

We have carefully considered all comments from the reviewers and revised our manuscript accordingly. The manuscript has also been double-checked, and the typos and grammar errors we found have been corrected. In the following section, we summarize our responses to each comment from the reviewers. We believe that our responses have well addressed all concerns from the reviewers. We hope our revised manuscript can be accepted for publication.

Case1

page 6, line 144: first µF is used, then uF

Response:
Thank for your comments. We have checked all units and replaced them.

Case2

Figure9: space missing. In the figure, the numbers are overlapping and the quality is bad. IL1 and IL2 must be written as indices.

Response:
Thank for your comments. We have checked all graphics and made changes.

Case3

Indices are not used in the complete text and the graphs. On the other hand, indices are mostly correct in the equations. This makes the manuscript hard to read.

Response:
Thank for your comments. We have checked all formulas and texts and made changes.

Case4

Also many equations look strange, e.g. Eq. (8): there is no "=" in this equation. Also brackets are not set in a proper way. The [ ] is too small in Eq. (8), just as an example. If the manuscript is written in latex, you should use \left[ instead of [.

Response:
Thank for your comments. We have checked all formulas and made changes.

Reviewer 3 Report

All concerns have been addressed well.

Author Response

Respose to the Review Comments

We would like to thank you for your careful reading, helpful comments, and constructive suggestions, which has significantly improved the presentation of our manuscript.

We have carefully considered all comments from the reviewers and revised our manuscript accordingly. The manuscript has also been double-checked, and the typos and grammar errors we found have been corrected. In the following section, we summarize our responses to each comment from the reviewers. We believe that our responses have well addressed all concerns from the reviewers. We hope our revised manuscript can be accepted for publication.

This paper mainly studied a two-way phase shift topology for wireless power transfer with reduced ripples.  The content and results are both solid. Some minor concerns are listed as follows.

Response:

We thank the reviewer for reading our paper carefully and giving the above positive comments. Thank you for your affirmation to us, and wish you a happy life and a smooth work.

Round 3

Reviewer 2 Report

The manuscript looks fine now.